# Corrosion Resistance and Biological Properties of Pure Magnesium Modified by PEO in Alkaline Phosphate Solutions

Mónica Echeverry-Rendón [1,2,*,†], Luisa F. Berrio [1,†], Sara M. Robledo [3], Jorge A. Calderón [1], Juan G. Castaño [1] and Felix Echeverría [1]

1   Centro de Investigación, Innovación y Desarrollo de Materiales CIDEMAT, Facultad de Ingeniería, Universidad de Antioquia UdeA, Calle 70 No. 52-21, Medellín 050010, Colombia
2   IMDEA Materials Institute, C/Eric Kandel 2, Getafe, 28906 Madrid, Spain
3   Programa de Estudio y Control de Enfermedades Tropicales PECET, Instituto de Investigaciones Médicas, Facultad de Medicina, Universidad de Antioquia UdeA, Calle 70 No. 52-21, Medellín 050010, Colombia
*   Correspondence: monica.echeverry@imdea.org; Tel.: +34-91549-3422
†   These authors contributed equally to this work.

**Abstract:** Magnesium (Mg) has been explored during the last few decades in the biomedical industry as a biodegradable implant. However, mechanical properties and corrosion resistance are still big concerns for clinical use. Therefore, this study proposes a suitable surface modification of the Mg by plasma electrolytic oxidation (PEO) to improve its corrosion resistance and biological performance. Mg samples were processed in a galvanostatic mode using an electrolytic solution of a phosphate compound supplemented with either potassium pyrophosphate or sodium-potassium tartrate. The obtained coatings were physiochemically characterized by SEM, XRD, EDS, and micro-Raman spectroscopy. The corrosion resistance of the coatings was studied using a hydrogen evolution setup and electrochemical tests. Finally, the biological performance of the material was evaluated by using an indirect test with osteoblasts. Obtained coatings showed a porous morphology with thicknesses ranging from 2 to 3 μm, which was closely dependent on the PEO solution. The corrosion resistance tests improved the degradation rate compared to the raw material. Additionally, an unreported active–passive corrosion behavior was evidence of a protective layer of corrosion products underneath the anodic coating. Indirect in vitro cytotoxicity assays indicated that the coatings improved the biocompatibility of the material. In conclusion, it was found that the produced coatings from this study not only lead to material protection but also improve the biological performance of the material and ensure cell survival, indicating that this could be a potential material used for bone implants.

**Keywords:** magnesium; degradable material; potassium pyrophosphate; sodium-potassium tartrate; plasma electrolytic oxidation

## 1. Introduction

Magnesium (Mg) is a biocompatible and bioabsorbable material that has been extensively explored during the last few decades as a biomedical implant. Due to its similarities with the mechanical properties of the bone, Mg seems to be a promising candidate for orthopedic implants [1–4]. However, the use of Mg is limited due to its high reactivity in aqueous media, in which $Mg(OH)_2$ and hydrogen gas ($H_2$) are produced in such a fast way that the body is not able to control them [5]. The release of high levels of $OH^-$ ions in a biological environment increases the pH, affecting the survival of the cells, considering that the physiological pH range of the body is between 7.4 and 7.7. On the other side, the accumulation of $H_2$ in the implant-material interface produces gas pockets or cavities, affecting the osseointegration process and leading to necrosis in the surrounding tissue [6,7].

The biocompatibility and mechanical performances of the Mg can be improved by different strategies, such as alloying elements or coatings [2,5,8]. Regarding Mg-alloys, some of the most popular are WE43, AZ31, and AZ91, which are designed in principle

for industrial purposes [8–10]. The biological validation of those materials has reported cytotoxic restrictions regarding some alloying elements. For instance, aluminum (Al) and rare earth (RE), despite promoting mechanical and corrosion resistance in Mg alloys, are considered toxic [11]. Al may induce neurological disorders [12], while RE leads to severe hepatotoxicity [13]. New alloys, including compatible elements such as Zn, Ca, and Sr, have been explored [14–16] to obtain an optimal alloy that combines good mechanical properties with high biocompatibility while decreasing the side effects during the degradation process of the implant. On the other side, coatings for Mg can be produced by two strategies: by deposition of new material on the surface or by growing the coating in situ. In this last group, plasma electrolytic oxidation (PEO) (also called micro-arc oxidation (MAO)) stands out as a simple, low-cost, reproducible, and effective technique.

The process consists of the oxidation of the material in a controlled and uniform way, generating a protective coating that improves the corrosion resistance and biocompatibility of the material [17]. In the PEO process, work parameters such as current density, voltage, time, and electrolytic solution can be modified and determine the final properties of the coating in aspects such as thickness, chemistry, and morphology [18,19]. The addition of elements and different compounds to the electrolytic solution plays an essential role in the composition and physicochemical characteristics of the coatings. Some of the most reported compounds used for Mg coatings are based on $SiO_3^{2-}$, $PO_4^{3-}$, and $AlO_2^-$ ions [19]. A phosphate-base electrolyte seems the best choice for orthopedic implants since phosphate species are essential for bone regeneration. This component is essential for hydroxyapatite deposition [20]. As part of the Mg coating, it improves its mechanical and corrosion resistance properties [5,21]. Even though phosphate base solutions have been extensively used for PEO treatment of Mg-alloys [17,21], only a few studies reported coatings formed on commercially pure magnesium (c.p. Mg). To produce a material with potential use for bone applications and avoid using alloys, this study aimed to produce a new coating produced on c.p. Mg by the PEO technique. A new formulation was tested, consisting of a base solution of sodium phosphate supplemented with either potassium pyrophosphate or sodium-potassium tartrate to improve the corrosion resistance and biocompatibility of the material.

## 2. Materials and Methods

### 2.1. Sample Preparation

Square samples of 1 cm × 1 cm × 0.1 cm of c.p. Mg (99.9%) were grounded with SiC papers up to grit 2000. Afterward, samples were cleaned in acetone and distilled water in an ultrasound bath for 15 min in each solution. The PEO process was carried out in a power supply (Matssusada Vol. 500-20, Charlotte, NC, USA). A stainless steel beaker was used as a cathode and the sample as the anode for the set-up. The base electrolytic solution consisted of $Na_3PO_4$ and NaOH, supplemented with either potassium pyrophosphate ($K_4P_2O_7$) or sodium potassium tartrate ($KNaC_4H_4O_6$). The system was immersed in a cooling water bath to control the evaporation of the solution. Samples were anodized in galvanostatic mode at 135 mA·cm$^{-2}$ for 600 s. A summary of the working conditions is presented in Table 1.

**Table 1.** Conditions used to obtain the PEO coatings on c.p. Mg.

| Sample Code | Formulation |
|---|---|
| M1 | 10 g/L $Na_3PO_4$–1 g/L NaOH |
| M2 | 10 g/L $Na_3PO_4$–1 g/L NaOH–10 g/L $K_4P_2O_7$ |
| M3 | 10 g/L $Na_3PO_4$–1 g/L NaOH–1 g/L $KNaC_4H_4O_6$ |

### 2.2. Material Characterization

Analysis of the chemical composition of the three coatings was performed using energy dispersive X-ray spectroscopy-EDS (OXFORD INCAPentaFET-x3, Abingdon, U.K). X-ray

diffraction (XRD) analyses were realized in Empyrean PANalytical equipment with Cu K$\alpha$ anode radiation (0.1541874 nm). The data were collected in the 2θ range from 10° to 100° at a scanning rate of 0.03° min$^{-1}$. Raman spectroscopy analyses were performed in a Micro-Raman Jovin Yvon Horiba, Model Labram High-Resolution Spectrometer, equipped with a confocal microscope Nikon BX41 (Tokyo, Japan), through a 50× objective, with a laser of He/Ne (632.5 nm). Raman spectra were registered with a pinhole of 1000 μm, a slit of 200 μm, and a scanning range from 100 to 2000 cm$^{-1}$. In addition, the surface morphology and thickness of anodized samples were observed by a scanning electron microscope (SEM) (JEOL JSM 6490 LV, Peabody, Massachusetts, USA). Samples were mounted in resin and then polished with 1 μm alumina powder to a mirror finish to analyze the cross-sections of the coatings.

### 2.3. Corrosion Tests

#### 2.3.1. Immersion Test

Samples were immersed in simulated body fluid (SBF) prepared based on the Kokubo guidelines [22] and maintained at 37 °C for five days in a water bath system. The surface and cross-section of the samples were studied with SEM (JEOL JSM 6490 LV) and EDS (OXFORD INCAPentaFET-x3). Additionally, the degradation rate of the samples was calculated based on a hydrogen evolution test conducted for 90 days in a 0.9%w/v NaCl (reagent grade, Merck, Rahway, NJ, USA) aqueous solution at 22 ± 3 °C. For comparison, a sample of c.p. Mg was included. The hydrogen evolution test was performed by suspending the samples using a fishing line to avoid the sample surfaces contacting the walls of the glass beaker used to contain the sample. As widely reported in the literature, a glass funnel and a burette were employed for hydrogen collection and volume measurement. Temperature and atmospheric pressure were 22 ± 3 °C and 0.85 bar, respectively. The amount of evolved hydrogen was measured daily in triplicate, with the average values presented in the results.

#### 2.3.2. Electrochemical Test

A potentiodynamic polarization test was performed using an Autolab PGSTAT 302 potentiostat, starting from the open circuit potential (OCP) with a scan rate of 0.33 mV/s. Measurements were made using an Ag/AgCl KCl 3 mol L$^{-1}$ ($E_{SHE}$ = 209 Mv) reference electrode in a solution of 0.9%w/v NaCl (reagent grade, Merck) at room temperature. A platinum mesh was used as a counter electrode, and the test area was about 0.78 cm$^2$.

### 2.4. Cytotoxicity Tests

The toxicity of the material was evaluated by an indirect test. Extracts of the samples were obtained from the immersion of three different samples with dimensions of 1 cm × 1 cm × 0.1 cm in 1 mL of McCoy's 5A Medium (Invitrogen, Waltham, MA, USA) supplemented with 10% of fetal bovine serum (FBS) (Invitrogen) for 24 h. On the other side, osteoblasts from the cell line Saos-2 (ATCC® HTB-85™, Virginia, USA) were seeded at a concentration of 10.000 cells/cm in a 96-well plate and incubated at 37 °C with 5% $CO_2$ and 95% relative humidity for 24 h. Then, the culture medium was removed and replaced with extracts diluted at different concentrations (100%, 50%, 25%, 12.5%, 6.25%, 3.12%, 1.56%, and 0.78%) and incubated at culture conditions for another 72 h. After that, the mitochondrial activity was measured by adding thiazolyl blue tetrazolium bromide (MTT) (Sigma, Cream Ridge, NJ, USA) at a concentration of 5 mg/mL in a ratio of 10 μL for each 100 μL of culture medium. Cells were incubated for 3 h, then the medium was removed, and the crystals of formazan produced were dissolved with DMSO. Finally, the optical density was measured at 570 nm in a spectrophotometer. The mitochondrial activity was measured after normalizing the values and compared to the control cells without any extracts. Each condition was evaluated in triplicate. The average and standard deviation were calculated and plotted using GraphPad Prism Version 7.03.3.

## 3. Results

### 3.1. PEO Process

Voltage-time curves of the PEO process for each sample are shown in Figure 1. Three distinct regions are observed: the first stage (i) shows a rapid rise in voltage where a compact layer begins; during this stage, the films grow following a linear relationship between voltage and treatment time, forming a barrier film, and ionic processes mostly conduct the charge. In the second stage (ii), a maximum voltage is reached, starting the discharge of sparks on the surface and producing a continuous process of melting/oxidation of the material; at this stage, the linearity of V vs. t is lost and the efficiency of the process gradually decays as the electron current component of the total current progressively increases. The maximum voltage registers for each solution were 276.8 V, 227.7 V, and 279.5 V for M1, M2, and M3, respectively. At this point, the crystallinity of the film occurs. Finally, in the third stage (iii), the voltage stabilizes, the sparks grow in size over time, the film thickness reaches its maximum value, and continuous breakdown events take place, leading to the final morphology of the PEO film and a largely increased proportion of electron current in the total current of the process. Rapid voltage fluctuations in stage III are related to the breakdown and healing of the coating during spark occurrence; more significant fluctuations are observed for M3, followed by M1 and M2. Further details on the PEO mechanism can be found elsewhere [17].

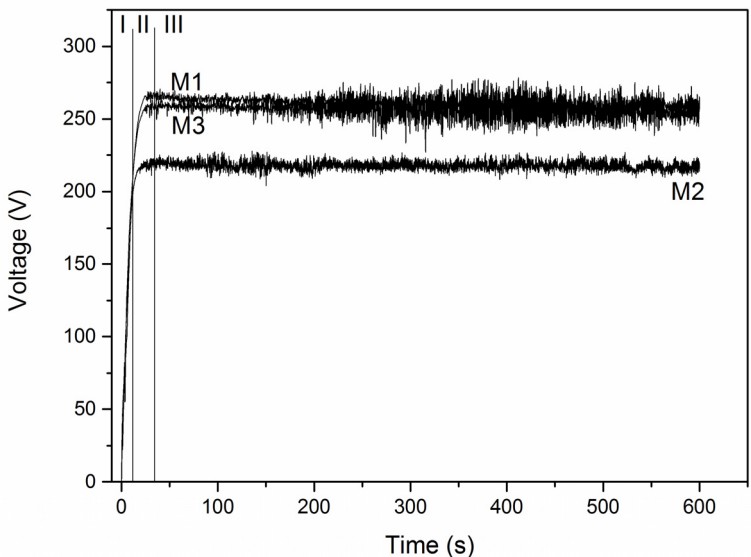

**Figure 1.** Voltage-time curves for each sample describe the three characteristics phases of the PEO process.

### 3.2. Characterization of the PEO Coatings

The composition, surface morphology, porosity, and thickness of the coatings are highly influenced by the species present in the PEO electrolyte. SEM images of surfaces and cross-sections of the PEO coatings obtained are shown in Figure 2. All the samples showed a porous morphology, characteristic of coatings obtained by PEO. The coatings of sample M1 showed a lower porous density than those of samples M2 and M3, which, oppositely, showed the most homogeneous morphology. M1 was anodized with the base electrolytic solution without additives, and the resulting coating showed a surface with several superficial cracks and circular pores of different sizes (Figure 2a). The cross-section of this sample (Figure 2d) shows a low internal porosity.

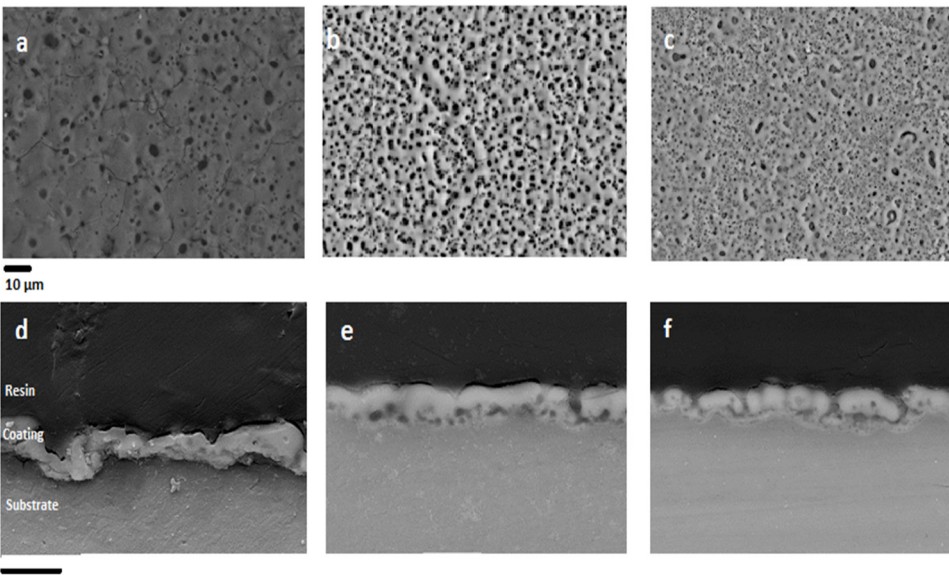

**Figure 2.** SEM images of the PEO coatings surfaces and their respective cross-sections: M1 (**a**,**d**), M2 (**b**,**e**), and M3 (**c**,**f**).

On the other hand, M2 coating, which adds $K_4P_2O_7$, results in a more homogeneous surface with pores with lower dimensions (Figure 2b). The cross-section of this sample shows a porous coating in which most of the pores are concentrated in the inner part of the coating close to the barrier layer and a more compact structure is observed on the outer region. In the case of M3 (Figure 2c), adding $KNaC_4H_4O_6$ leads to even smaller porosity at the surface with some big porous structures in a random distribution. In the corresponding cross-section SEM image (Figure 2f), the internal morphology of the coating is similar to that of M2; however, some passing through pores was observed. The thickness dimensions for all the coatings were similar, ranging from about 2 to 3 $\mu$m.

The chemical compositions of all the coatings are presented in Table 2. The incorporation of phosphorus and sodium from the PEO solution was found in all coatings but in a lower percentage in the M3 sample. Only in M2 was the presence of potassium (K) detected.

**Table 2.** EDS chemical composition of the PEO coatings (wt%). Not including Mg and O.

|     | P   | Na  | K   |
| --- | --- | --- | --- |
| M1  | 8.3 | 0.9 |     |
| M2  | 8.2 | 1.5 | 1.0 |
| M3  | 5.7 | 0.4 |     |

XRD results are shown in Figure 3. This analysis evidenced the formation of $Mg_3(PO_4)_2$ in the three coatings with characteristic peaks of high intensity, varying with the PEO electrolyte. The Raman results in Figure 4 clearly show magnesium oxide (MgO) peaks in all three samples. The characteristic bands are located at 1500 and 2000 cm$^{-1}$. Another phase observed was $Mg_3(PO_4)_2$, with bands at 1000 and 1200 cm$^{-1}$, respectively. Additionally, bands at 270 and 450 cm$^{-1}$ indicated that $Mg(OH)_2$ was detected. According to the Raman peaks intensity, there is no apparent difference in the three species between the samples.

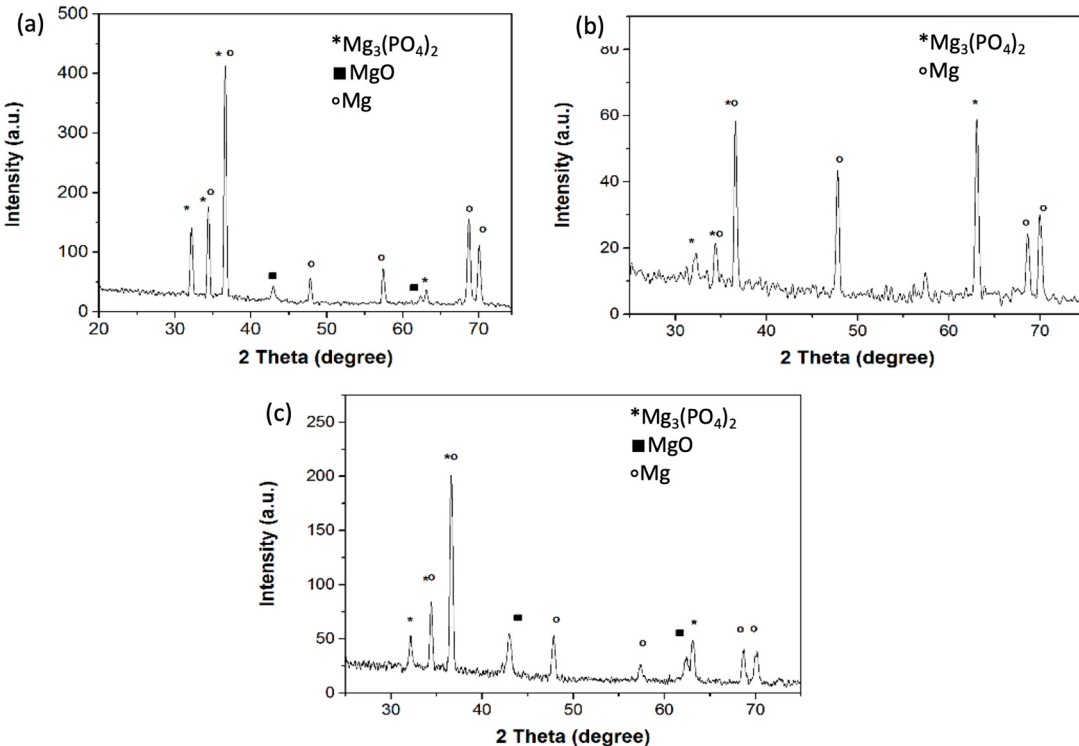

**Figure 3.** XRD diffractograms of the PEO coatings for the samples: M1 (**a**), M2 (**b**), and M3 (**c**).

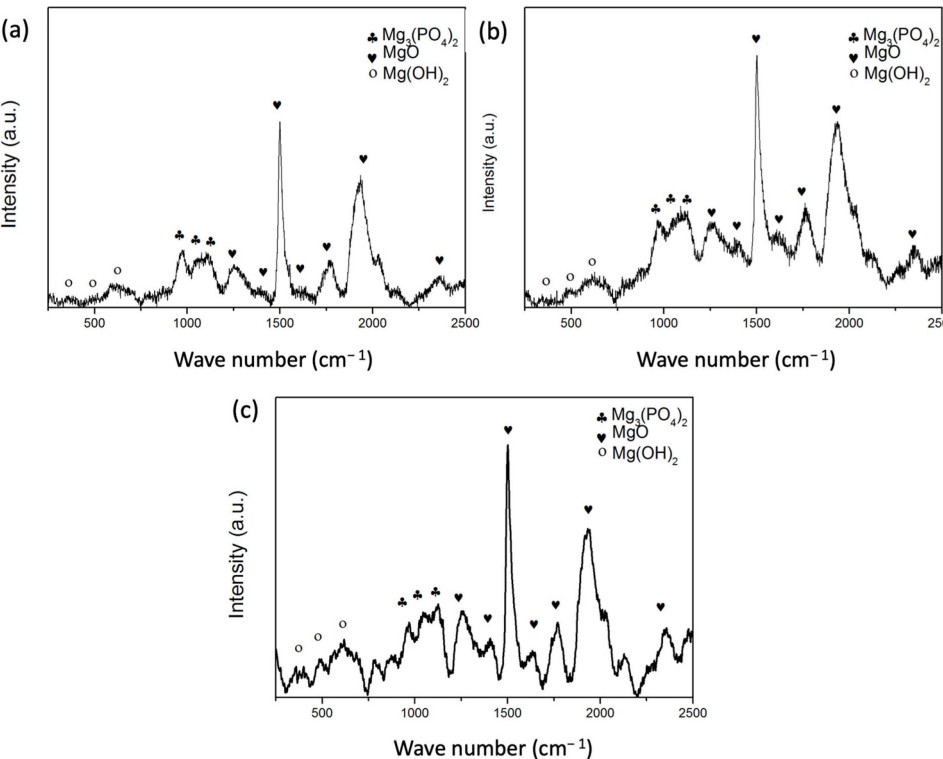

**Figure 4.** Raman spectra of the PEO coatings for the samples: M1 (**a**), M2 (**b**), and M3 (**c**).

*3.3. Corrosion Tests*

3.3.1. Immersion Test

After 5 days of immersion in SBF, the chemical composition of the surface samples was studied by EDS analysis, and the results are summarized in Table 3. According to

this, calcium (Ca) and chlorine (Cl) became part of the coating, and the concentration of phosphate (P) increased for all the samples, suggesting the formation of calcium phosphate compounds.

**Table 3.** Chemical composition of the coatings after the immersion test (wt%). Not including Mg and O.

|     | P    | Na  | K   | Ca  | Cl  |
| --- | ---- | --- | --- | --- | --- |
| M1  | 9.9  | 0.5 | -   | 2.6 | 0.3 |
| M2  | 25.1 | 1.7 | 0.8 | 3.7 | 0.4 |
| M3  | 15.0 | 1.9 | -   | 5.2 | 0.5 |

Images of the surface and cross-section of the coated samples after immersion for 5 days in SBF are shown in Figure 5. Comparing with the samples before the immersion, it is observed that significant changes took place in both surface morphology and porosity. Additionally, the cross-sections of the samples after immersion (Figure 5d–f) show that the coatings reacted with the electrolyte, resulting in variations in both internal morphology and thickness of the coating. However, despite all these changes, the coatings are still present in all cases. Moreover, the corrosion process is evidenced by the changes in the coatings and the reaction of the substrate, as evidenced by the layer of corrosion products observed underneath the coating; this layer appears to be more abundant for M1 (Figure 5d).

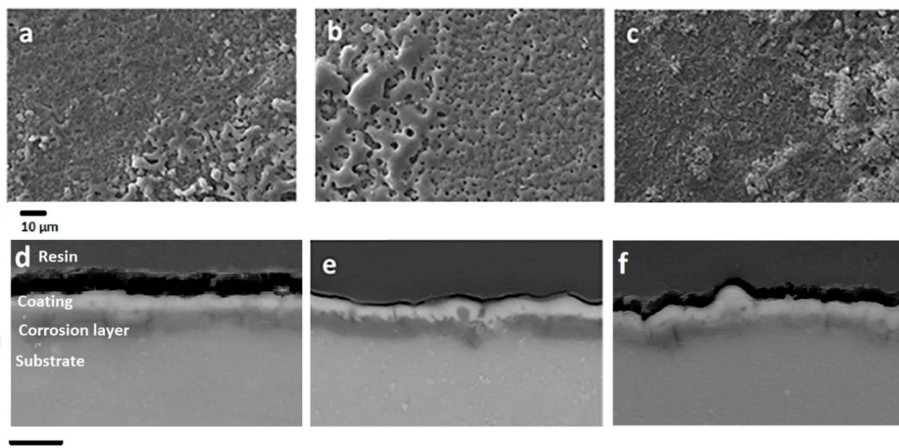

**Figure 5.** SEM micrographs of the surface of 3 Mg PEO coated samples after 5 days of immersion in SBF solution: (**a**) M1, (**b**) M2, and (**c**) M3. Cross-section SEM micrographs after 5 days of immersion in SBF solution: (**d**) M1, (**e**) M2, and (**f**) M3.

Figures 6 and 7 show the XRD and Raman results for all samples after the immersion test. It is possible to observe that calcium phosphate was present in all the coatings.

The phosphate species from the electrolyte react with $Mg^{2+}$ ions from the substrate to form $Mg_3(PO_4)_2$, as confirmed by XRD, Raman, and EDS results (Figures 6 and 7, and Table 3, respectively).

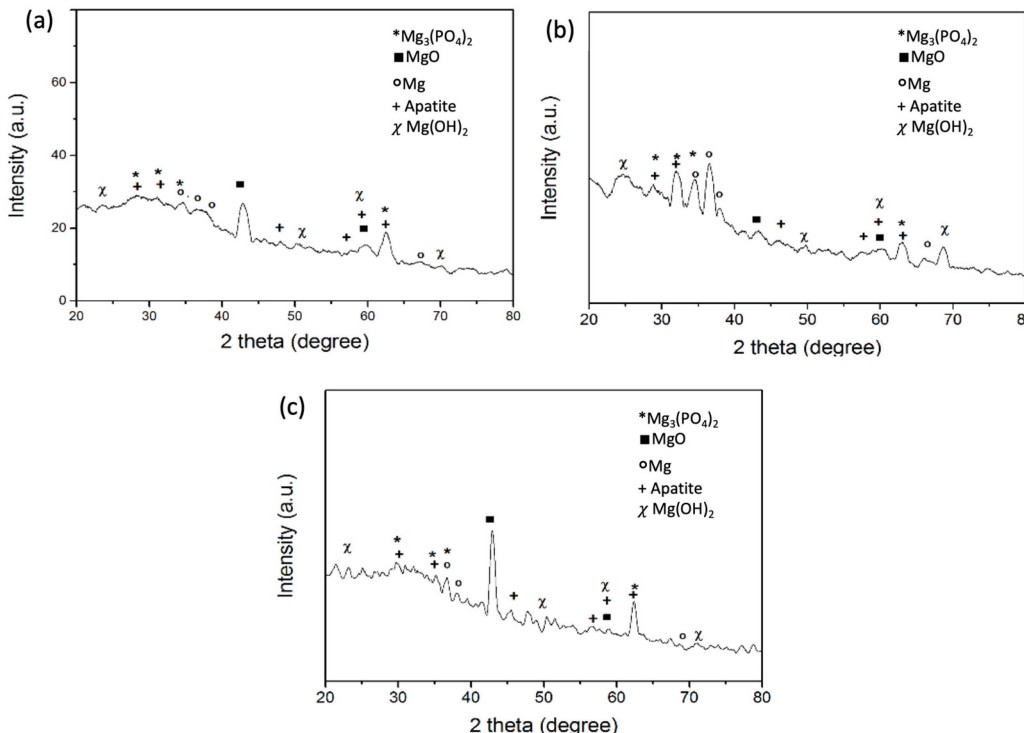

**Figure 6.** XRD diffractograms of 3 Mg PEO coated samples after 5 days of immersion in SBF solution: (**a**) M1, (**b**) M2, and (**c**) M3.

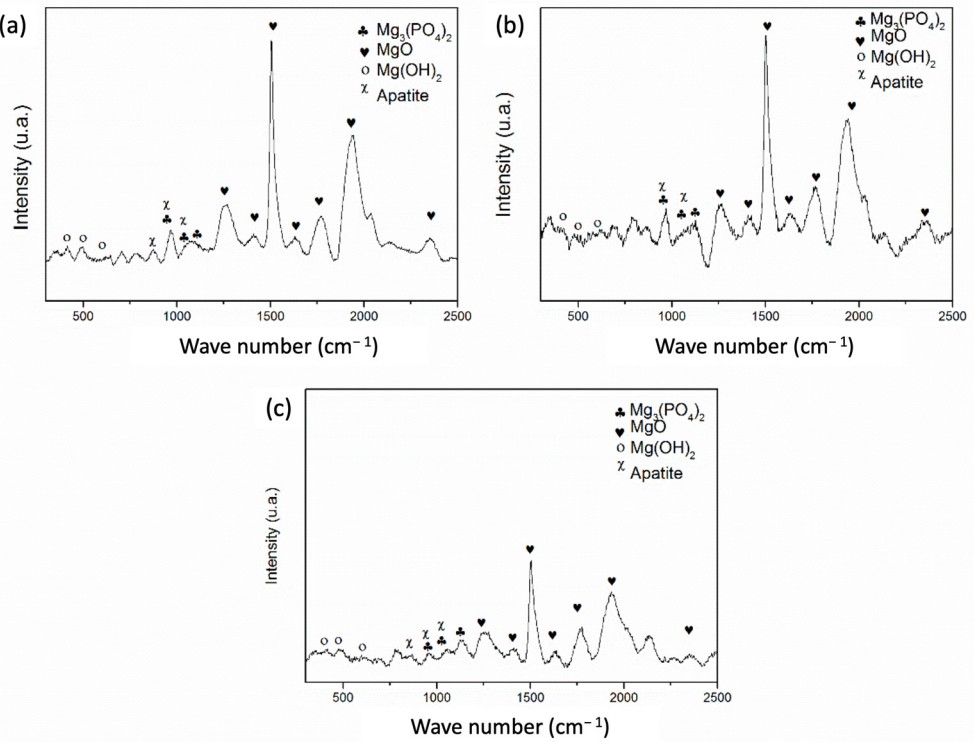

**Figure 7.** Raman spectra of 3 Mg PEO coated samples after 5 days of immersion in SBF solution: (**a**) M1, (**b**) M2, and (**c**) M3.

### 3.3.2. Hydrogen Evolution

The degradation of Mg is directly related to hydrogen production, following the reactions below and resulting in Equation (1):

$$Mg \rightarrow Mg^{2+} + 2e^-$$

$$2H_2O \rightarrow 2(OH)^- + 2H^+$$

$$2H^+ + 2e^- \rightarrow H_2 \uparrow$$

$$Mg^{2+} + 2(OH)^- \rightarrow Mg(OH)_2$$

$$Mg + 2H_2O \rightarrow Mg(OH)_2 + H_2 \uparrow \tag{1}$$

Based on this, the degradation rate of the Mg samples was calculated using the results from the hydrogen evolution experiment and applying Equation (2).

$$P_H = 2.279 \, V_H \tag{2}$$

where the constant 2.279 is employed to calculate the corrosion rate $P_H$ in mm·year$^{-1}$ when the volume of evolved hydrogen $V_H$ is expressed in ml·cm$^{-2}$·d$^{-1}$, as reported in the literature [23].

The corrosion rate in terms of hydrogen evolution as a function of time is shown in Figure 8. All samples exhibited active–passive corrosion behavior. An initial rapid increase in corrosion rate reached a maximum at around 15 days of immersion, which was higher for the c.p. Mg followed by M1, M2, and M3. After that, the corrosion rate of the samples decreased. A reduction in the corrosion rate indicates passivation processes; however, this occurs differently for the three coatings. In M2 and M3, it took a couple of weeks, whereas for M1, passivation seems to start only after about three or four weeks of immersion. After about 30 days, the corrosion rate for M1 again rose with a steep slope until about 80 days, when it appeared to reach a maximum. For M2, the corrosion rate also decreased until approximately 30 days; after that, the behavior remained almost constant until the end of the test. In the case of M3, after a slight descent in corrosion rate at around 30 days, it was nearly constant for about 15 days more, with a small increment at the end of the test.

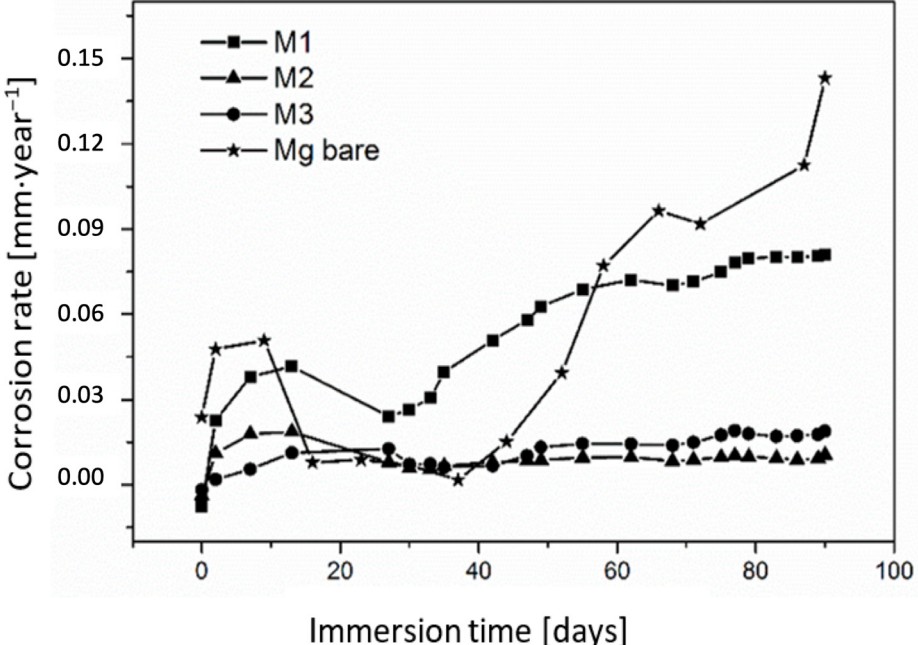

**Figure 8.** Corrosion rate calculated by the hydrogen evolution test of bare and PEO coated Mg samples immersed in 0.9%w/v NaCl aqueous solution.

### 3.3.3. Electrochemical Test

Potentiodynamic polarization tests were conducted on bare and coated Mg samples to determine their electrochemical behavior in a saline solution. Figure 9 shows similar behavior for all samples, confirming the existence of active–passive behavior, as was previously observed in the hydrogen evolution test. Table 4 shows the corrosion potential ($E_{corr}$), corrosion current density ($i_{corr}$), passivation potential ($E_{pp}$), passivation current density ($i_{pass}$), pitting potential ($E_{pit}$), and instantaneous corrosion rates calculated by polarization measurements (CR, Pi). The $E_{corr}$ moves towards more positive values, going from more negative values for M1, followed by M2, and finally, the more positive M3. All these values are more negative than the experimental value obtained for c.p. Mg, which was about −1.62 V. The Tafel extrapolation method from the cathodic branch of the polarization curves found the corrosion current densities obtained from polarization curves. The cathodic Tafel slopes observed at the polarization curves exhibited values of around 280 mV·dec$^{-1}$.

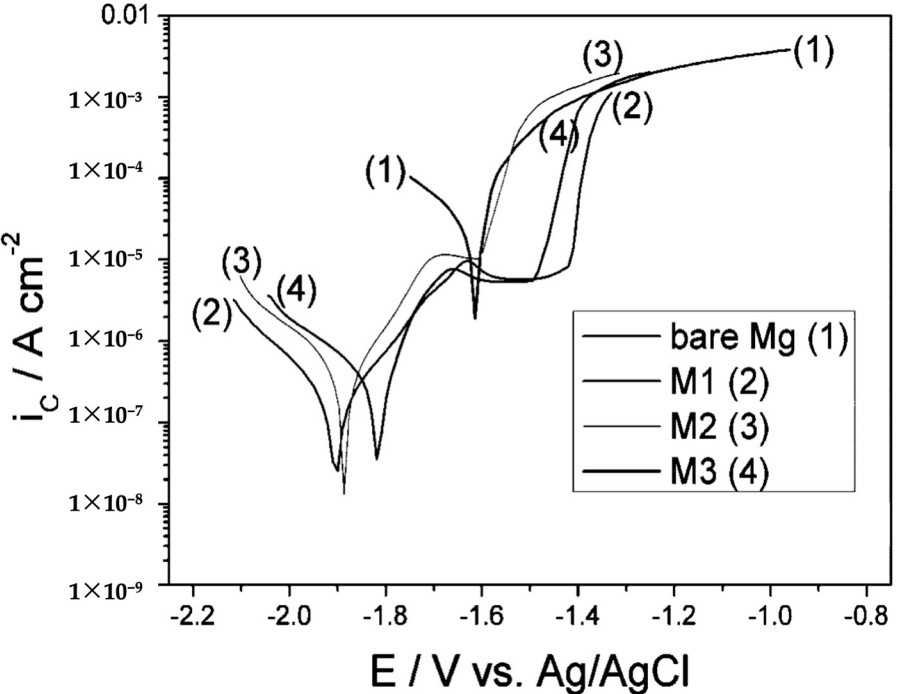

**Figure 9.** Potentiodynamic polarization test of 3 Mg PEO coated samples in 0.9%w/v NaCl aqueous solution.

**Table 4.** Polarization curve parameters for bare Mg and PEO coated samples in 0.9%w/v NaCl aqueous solution.

| Sample | $E_{corr}$ [V] | $i_{corr}$ [A·cm$^{-2}$] | $E_{pp}$ [V] | $i_{pass}$ [A·cm$^{-2}$] | $E_{pit}$ [V] | $E_{pit} - E_{corr}$ [V] | CR, $P_i$ [mm·year$^{-1}$] |
|---|---|---|---|---|---|---|---|
| c.p Mg | −1.62 | $2.6 \times 10^{-5}$ | - | - | - | - | 0.5900 |
| M1 | −1.90 | $1.60 \times 10^{-7}$ | −1.62 | $5.64 \times 10^{-6}$ | −1.42 | 0.48 | 0.0036 |
| M2 | −1.89 | $4.22 \times 10^{-7}$ | −1.69 | $1.05 \times 10^{-5}$ | −1.61 | 0.28 | 0.0096 |
| M3 | −1.82 | $3.12 \times 10^{-7}$ | −1.65 | $5.33 \times 10^{-6}$ | −1.50 | 0.32 | 0.0071 |

By contrast, the coated samples exhibited significantly lower corrosion current densities. The corrosion current density of Mg was decreasing because the barrier action of the coatings was around two magnitude orders. Compared to this, the reduction of the corrosion rates in the coated samples taken from hydrogen evolution was just one order of magnitude lower than in the bare sample. The lower the $I_{corr}$, the higher the corrosion resistance, which can be related to forming protective barriers [24]. The polarization curves

in Figure 9 for the coated samples show the occurrence of a passivation process in all anodized samples. Although passivation occurs at about the same potential, the pitting potential was more positive for the M1 sample, followed closely by the M3 and M2 samples, which showed the more negative value of the three coated samples. The pitting resistance, measured as the difference between the pitting and corrosion potentials, was higher for the sample coated in the base electrolyte (M1). In contrast, the samples processed in the M3 and M2 electrolytes presented similar lower values. M2 showed a higher passivation current, whereas M1 and M3 samples showed similar values.

### 3.4. Cytotoxicity Tests

Because of the high interference between Mg and the elements used to reveal the system, the indirect method used for cytotoxicity tests only considered lower dilutions [25]. For extract concentrations of 6.25% and above, cell viability was below 30%. Below a concentration of 6.25%, the viability of cells was high, and samples showed a non-toxic effect for the 3 anodized samples, as seen in Figure 10. With the decrease in Mg concentration, the number of dead cells also decreased. These results indicate similar behavior for the three coatings. No statistically significant differences among samples were observed ($p > 0.05$).

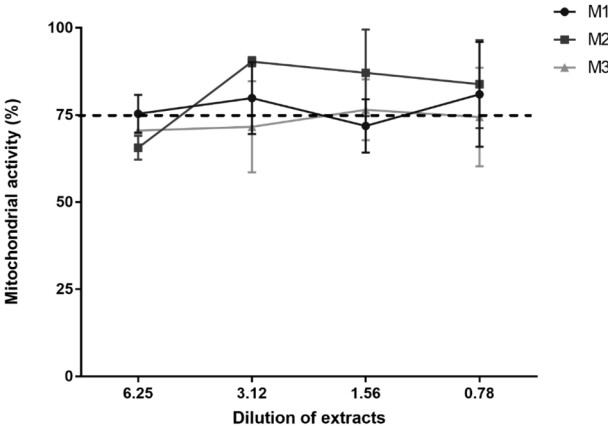

**Figure 10.** Results of the cytotoxicity test for the PEO treated samples: M1, M2, and M3.

## 4. Discussion

The results obtained in the present study propose two new formulations to improve the corrosion and biological performance of the Mg. The coating formation by using the additives potassium pyrophosphate ($K_4P_2O_7$) and sodium potassium tartrate ($KNaC_4H_4O_6$) consisted of the three main steps usually reported in several previous studies and illustrated in the anodization curves (Figure 1) [26]. The obtained coatings presented the characteristic morphology of PEO coatings (Figure 2) [27], in which pores of different sizes and shapes are observed across the surface. The PEO process on Mg alloys induces the incorporation of elements from the electrolyte, resulting in different compounds [26,28]. In the present case, in addition to MgO, $Mg_3(PO_4)_2$ was formed during the PEO process. Coatings containing these two compounds have been reported to be protective against corrosion [29,30]. Variations in internal porosity and thickness can be produced depending on the additive in the electrolytic solution. The form, distribution, and size of these pores are related to the sparks generated, which are also linked with the conductivity and nature of the electrolyte [26]. These effects are also evident in the cross-sections of the coatings (Figure 2), where variations in internal porosity and thickness are revealed. Analysis of the variations observed between the three samples studied here suggests that the more distinctive characteristic was the porosity of the PEO films, as little or no differences were detected in either chemical/microstructural composition (evidenced by EDS, XRD, and Raman results) or thickness. The surface morphology, which is the film porosity observed in Figure 2 (number and size of pores), appears to have an inverse relationship with the

corrosion resistance of the film, while the number and size of the passing pores present a direct relationship with the corrosiveness of the sample. The corrosion resistance of the samples has this order: M2 > M3 > M1; for the surface porosity: M2 > M3 > M1; and the passing porosity: M1 > M3 > M2. Therefore, it is clear that the additives modified the electrolyte's physical properties [31,32], affecting the morphology of the resulting PEO films. The observed morphologies agree well with the models proposed for PEO films on Mg alloys [17], with a porous surface layer followed by a more compact one, and finally, towards the PEO film/substrate interface, a thin barrier layer.

The morphological characteristics of the coatings allow for regulating the corrosion rate. Immersion of the coated samples and, consequently, the formation of corrosion products induce a reduction in the porosity, as suggested by the changes in morphology at the surface and in the cross-sections (Figure 5). During the first hours of immersion, the high hydrogen evolution indicates that the electrolyte initially reaches the metallic Mg substrate through the coating pores, reacting and forming corrosion products [33]. These products are mainly $Mg(OH)_2$ [34] and result in the reduction of the corrosion rate of the samples as the hydroxide Mg layer grows at the coating substrate interface. Consequently, the aggressive species have more difficulty reaching the substrate material, and the corrosion rates gradually decrease, as depicted by the corrosion curves in Figure 8. Physicochemical characterization after the immersion of the samples showed a presence of calcium phosphate that was not detected before; this compound could be of great interest for biomedical applications since, according to the literature, it promotes cell-material interactions [18,30,35].

Finally, as the $Mg(OH)_2$ layer continues thickening and blocking the entrance of aggressive species through the pores, the corrosion rate further decreases until reaching a more or less stable value after several days of immersion. This mechanism is similar to that proposed by other authors [36,37]. However, an active–passive behavior is proposed, which was also revealed by the potentiodynamic polarization curves of the anodized samples (Figure 9). In this manner, although the anodic coating does not fully protect the substrate, it improves the stability of the $Mg(OH)_2$ layer, which appears to be responsible for the reduction of the substrate corrosion rate (Figure 5). In addition, considering that the cross-sections in Figure 5 reveal the existence of a continuous and relatively thick corrosion product layer already after 5 days of immersion. However, Figure 8 indicates that the corrosion rate only starts decreasing after 10 to 15 days of immersion; it can be concluded that a critical thickness of the $Mg(OH)_2$ layer is required to induce a protective behavior of the surface. This thickness might be related to the characteristics of the anodic oxide film, as indicated by the differences of such a layer observed in Figure 5 and the variations in the curves in Figure 8. A later increase in the corrosion rate must be related to failures in the anodic oxide film, as proposed by Lin et al. [36]. Furthermore, the coating material is also attacked by the electrolyte, becoming thinner with time; however, the corrosion rate of the anodic layer appears to be very low, as it remains present on the surface all the time. This mechanism is depicted in Figure 11. The more evident effect of the additives on the mechanism of formation of PEO coatings described here is that the change in the physical properties of the electrolyte caused by the insertion of the additives (i.e., electrolytic conductivity) leads to a variation in the number of passing pores in the film, which directly affects the resistance to the entrance of aggressive ions and consequently the reaction of those ions with the Mg substrate. In other words, the introduction of the additives to the PEO electrolyte directly affects the corrosion resistance of the PEO film; this agrees with the results reported in Figure 5, where the layer of corrosion products formed underneath the PEO film appears to be thicker for M1. In addition, the reduction in the formation of passing pores is accompanied by an increment in surface porosity. Further work is required to establish if there is a relationship between the amount of additive and the corrosion resistance provided by the PEO treatment.

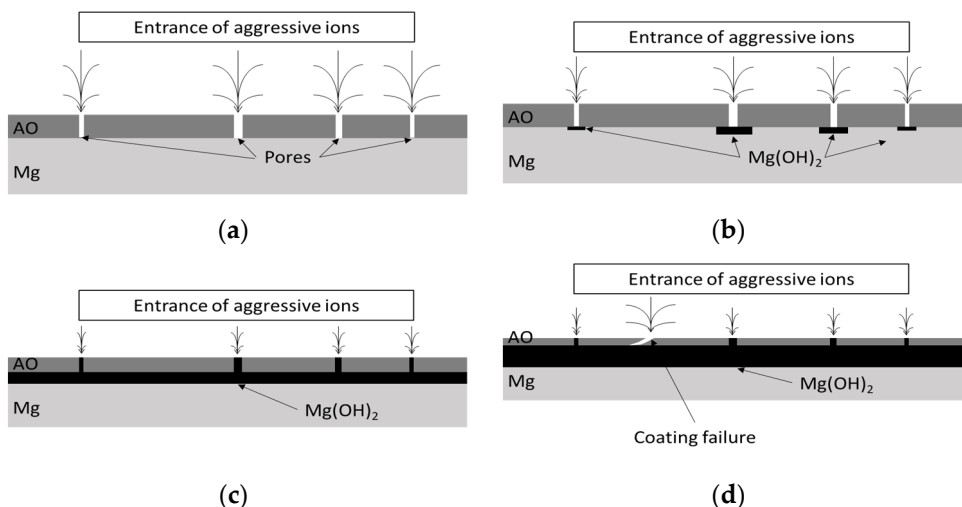

**Figure 11.** Illustration of the corrosion process of anodized c.p. Mg in an immersion test. (**a**) Once the sample is immersed, aggressive ions reach the substrate through the coating pores. (**b**) As a result of the reaction of the Mg substrate with the electrolyte ions, $Mg(OH)_2$ starts forming at the base of the pores, gradually reducing the high initial corrosion rate. (**c**) The corrosion process develops along the substrate/coating interface, blocking the diffusion of the aggressive ions, and the corrosion rate is stable. (**d**) Failure of the coating leads to an increase in the entrance of aggressive species and a rise in the corrosion rate of the substrate.

Generally, the corrosion rate of Mg alloys obtained from hydrogen evolution differs from other methods frequently used (for example, mass loss). This disagreement was explained by different authors [38,39], noting that those calculations did not consider the amount of hydrogen that remains dissolved in the electrolyte. In our experiments, the solution was pre-saturated with $H_2$ to solve the problems related to dissolved $H_2$. Other possible sources of error were negligible or easily avoided, including the variation of gas solubility with temperature and the polymeric material permeable to $H_2$ in the measuring setup.

According to the electrochemical test, the corrosion current densities were found by the Tafel extrapolation method from the cathodic branch of the polarization curves. This procedure is the most recommended due to the existence of a cathodic control of the corrosion process of Mg and taking into account the existence of an anomalous phenomenon called the "Negative Difference Effect (NDE)" during the anodic polarization [40–42]. The essential feature of the NDE is that the rates of both the anodic and cathodic reactions increase with the applied anodic potential, which is unusual [39]. Furthermore, the cathodic Tafel slopes observed at the polarization curves exhibited around 280 $mV \cdot dec^{-1}$, usually reported for cathodic hydrogen evolution related to water reduction on Mg [41,42]. The $i_{corr}$ value for the c.p. Mg sample was $2.6 \times 10^{-5}$ $A \cdot cm^{-2}$, which agrees with values reported in the literature [43].

The disagreement between the corrosion rates taken from hydrogen evolution and electrochemical measurements is broadly discussed in the literature, which is always related to the NDE anomalous phenomenon [40,41]. The corrosion products accumulate under the coating (Figure 5d–f), leading to a protective behavior as evidenced by a reducing and leveling hydrogen evolution rate (Figure 8). The transition between active and passive in the samples was evident in the coated samples. However, it was not so evident in uncoated Mg, as can be seen in the polarization curves (Figure 9). Thus, during the first hours of immersion, the pH value increases, and consequently, $Mg(OH)_2$ becomes stable and precipitates into the coating pores, protecting the substrate against further corrosion. The formation of $Mg_3(PO_4)_2$ might also contribute to the protection of the substrate, but to a lesser extent [44]. This mechanism is supported by observing a layer of corrosion products formed underneath the anodic coating, which is still in good condition after immersion

(Figure 5). Despite the corrosion reaction affecting all substrate surfaces, as evidenced by the continuous layer seen in Figure 5, the presence of the coating reduces the active surface area.

Although the electrochemical results indicate that in the short term, the additives do not affect the corrosion resistance of the coating, in the long term, both additives significantly increase the corrosion resistance of the anodic coating, as evidenced by the hydrogen evolution test. In potassium pyrophosphate, the corrosion rate becomes about eight times lower than the coating without any additive, whereas for sodium potassium tartrate, the reduction is around four times that of the base electrolyte. Looking at the curves in Figure 8, the corrosion rate during the first hours of immersion is similar for the coatings with and without additives; however, the slope of the curves gradually reduces during the first couple of weeks of immersion. After that, lower corrosion rates are observed in M2 and M3 samples. Analysis of the chemistry of the coatings revealed nearly no differences; only the XRD results indicated that the M2 coating had no content of crystalline MgO.

The better corrosion performance of M2 compared with M3 could be related to the porosity characteristics of the coatings. The sparks and, consequently, the pores formed in potassium pyrophosphate (M2) are different from those in sodium potassium tartrate (M3), as confirmed in the surface images in Figure 5. After the immersion test, $Mg(OH)_2$ is possibly formed on the coating surface due to the transformation of MgO [45], $MgCl_2$ formed from $Mg(OH)_2$ [46], and $Ca_3(PO_4)_2$ from the electrolyte (Figures 6 and 7).

The coatings formed on Mg in the present study seem to be a good option for producing Mg biomaterials. The in vitro assay results indicated that cells survived more than 75%, meaning these surfaces are non-toxic to humans [47]. Therefore, in addition to controlling the corrosion rate of Mg and, consequently, the mechanical integrity of the material; also, the survival of cells is assured. The latter could be explained as when the corrosion process of Mg is reduced and the formation of OH⁻ ions is controlled. Being $OH^-$ a byproduct of that reaction (Equation (1)), the evolution of $H_2$ is maintained at acceptable levels for the human body, and the pH of the medium is kept at values appropriate for cell survival [41]. However, bioactivity relies not only on the Mg substrate's corrosion rate but also on the elements present in the coating. It is well known that calcium-phosphate species are convenient for interacting with a biomaterial surface and the surrounding tissue.

On the other hand, it has been reported that the formation of calcium-phosphate compounds on the surface of a bare Mg alloy is inhibited by the formation of $Mg(OH)_2$ [48]. In the present study, anodization of pure Mg allows the formation of those Ca-P compounds during immersion in SBF, as evidenced by both XRD and Raman spectroscopy (Figures 9 and 10). This consequently will help the formation of bone tissue on the surface of the Mg biomaterial.

## 5. Conclusions

PEO treatment of c.p. Mg in alkaline aqueous solutions allows for obtaining relatively thick porous coatings with different morphological characteristics. Morphological changes are mainly related to the size and distribution of the coating pores, and these variations depend on the chemistry of the electrolyte employed. According to the results of the present work, the more effective additive of the PEO film obtained in the coating without any additive protected the substrate, the corrosion resistance of the PEO film obtained in the coating without any additive provided protection to the substrate, and the corrosion resistance of the PEO coatings obtained with additives was much better and also more stable with time. Both immersion and electrochemical testing of the Mg substrate/anodic coating system reveal active–passive corrosion behavior. The anodic layer induces passivation of the surface due to precipitation of $Mg(OH)_2$ and $Mg_3(PO_4)_2$, which form a corrosion products layer underneath the anodic coating when immersed in a medium containing Cl⁻ ions. As a result, corrosion rates are kept at much lower values than the bare Mg substrate. The anodic surface treatment in phosphate solutions, with and without

potassium pyrophosphate and sodium-potassium tartrate additives, protects the substrate and improves the biological activity of the material, ensuring cell survival.

**Author Contributions:** Conceptualization and methodology: M.E.-R. and L.F.B.; Formal Analysis, M.E.-R., L.F.B., S.M.R., J.A.C., J.G.C. and F.E.; writing—original draft preparation and editing, M.E.-R., L.F.B., J.A.C., J.G.C. and F.E.; Funding acquisition, F.E. All authors have read and agreed to the published version of the manuscript.

**Funding:** This study was funded under the project code 2014-657. Grant CODI-UdeA, and the grant 406-2020, Colombian Ministry of science.

**Data Availability Statement:** Data sharing is not applicable to this article.

**Conflicts of Interest:** The authors declare that they have no conflict of interest.

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
