# Peer review of "Corrosion Resistance and Biological Properties of Pure Magnesium Modified by PEO in Alkaline Phosphate Solutions"

_cmd, doi:10.3390/cmd4020012_

Round 1

Reviewer 1 Report

In this work, the influence of surface modification on the corrosion resistance and biological properties of magnesium was investigated. The subject of this study is very interesting and significant from a practical point of view, especially because the behavior of the samples was observed over a long period of time. However, in order to improve and clarify the whole manuscript, some parts of the work need further refinement.

1. The composition of the simulated body fluid (SBF) is not specified.

2. The settings for the experimental measurements are poor: the immersion test was performed in SBF (at 37°C for 5 days) and in 0.9%w NaCl (for 90 days, at what temperature?). After the immersion test in SBF, SEM/EDS, XRD and Raman analyzes of samples M1, M2, M3 were performed. Why were no corrosion tests performed (e.g., hydrogen measurements or polarization measurements)?

On the other hand, during immersion in 0.9%w NaCl, the corrosion rate was determined (by hydrogen measurement) and polarization measurements. What about the surface analysis? If these experimental conditions were reasonably harmonized, the discussion (Section 4) would probably be of better quality. The manuscript really has great potential.

3. The experimental section lacks a description of the method for hydrogen determination. Also, the conditions for hydrogen volume determination are not specified; they depend on temperature and pressure. In addition, Equation 2 is not clear; what does the constant 2.279 involve?

4. The discussion is scattered and not focused on the subject of the paper. It is too focused on the corrosion of pure Mg. For example, why is Figure 12 highlighted in this way? How would changing the surface with different layers change the condition in Figure 12? That is more important, and that is what is missing in this work. In general, this work lacks a quality comparison of the results obtained and how they relate to the data in the literature. In the end, it is not at all clear which surface layer M1, M2 or M3 has better corrosion properties.

5. Why is NDE important for the determination of icorr? NDE is the phenomenon of abnormal behavior of Mg and Al, where at high anodic potentials the dissolution of the metal is accompanied by the evolution of hydrogen. Abnormal behavior is also observed in these metals at high cathodic potentials: in addition to the evolution of hydrogen, the metal dissolves in parallel. Therefore, the NDE should have no effect on the quantities determined at the corrosion potential or at the open circuit potential.

6. As far as I know, a Tafel slope of about 120 mV/dec is characteristic for the hydrogen evolution reaction.

7. Line 282: see references [56].

8. Check that the references complies with journal regulations.

9. The details of the authors' contributions are missing.

Author Response

  1. The composition of the simulated body fluid (SBF) is not specified.

Answer: The SBF solution was prepared based on the Kokubo guidelines. In the manuscript, the ref was added. T. Kokubo, H. Takadama, How useful is SBF in predicting in vivo bone bioactivity?, 2006. Biomaterials, 27 (15), 2907-2915, https://doi.org/10.1016/j.biomaterials.2006.01.017.

  1. The settings for the experimental measurements are poor: the immersion test was performed in SBF (at 37°C for 5 days) and in 0.9%w NaCl (for 90 days, at what temperature?). After the immersion test in SBF, SEM/EDS, XRD and Raman analyzes of samples M1, M2, M3 were performed. Why were no corrosion tests performed (e.g., hydrogen measurements or polarization measurements)?On the other hand, during immersion in 0.9%w NaCl, the corrosion rate was determined (by hydrogen measurement) and polarization measurements. What about the surface analysis? If these experimental conditions were reasonably harmonized, the discussion (Section 4) would probably be of better quality. The manuscript really has great potential.

Answer: The test in 0.9%w NaCl was carried out a a temperature of 22 ± 3 °C. The correction will be included.

In line with the reviewer observation, the inmersion and corrosion testing in the NaCl solution, was done considering that the more agresive ion in the SBF solution was chloride, and therefore the authors expected, that using this concentration of NaCl, the corrosiveness of chloride in the two solutions was closely replicated, but more clearly analysed in the NaCl solution, in absence of the other components of SBF. On the other hand, surface analysis apperead to us more important in the samples inmersed in the SBF solution as the other components of this electrolyte could interact with the sample surfaces, forming compounds of biological interest, such as hidroxyapatatite. According to this, the authors consider the discussion section is still valid as the experimental conditions were in fact “harmonized“. In any case the discussion section was reviewed following the reviewer sugestion.

3. The experimental section lacks a description of the method for hydrogen determination. Also, the conditions for hydrogen volume determination are not specified; they depend on temperature and pressure. In addition, Equation 2 is not clear; what does the constant 2.279 involve?

Answer: The hydrogen evolution test was performed suspending the samples employing a fishing line, in order to avoid that the sample surfaces contact the walls of the glass beaker used to contain the sample. A glass funel and a burette were employed for hydrogen collection and volume measurement, as widely reported in the literature. Temperature and atmosferic pressure were 22 ± 3 °C and 0.85 bar, respectively. The amount ofevolved hydrogen was measured daily by triplicate, with the average values presented in the results. The constant 2.279 is employed to calculate the corrosion rate in mm·year-1 when the volume of evolved hydrogen is expresed in ml·cm-2·d-1 as reported by N. I. Zainal Abidin, D. Martin, and A. Atrens, “Corrosion of high purity Mg, AZ91, ZE41 and Mg2Zn0.2Mn in Hank’s solution at room temperature,” Corros. Sci., vol. 53, no. 3, pp. 862–872, Mar. 2011. This information will be included in the corrected version.

4. The discussion is scattered and not focused on the subject of the paper. It is too focused on the corrosion of pure Mg. For example, why is Figure 12 highlighted in this way? How would changing the surface with different layers change the condition in Figure 12? That is more important, and that is what is missing in this work. In general, this work lacks a quality comparison of the results obtained and how they relate to the data in the literature. In the end, it is not at all clear which surface layer M1, M2 or M3 has better corrosion properties.

Answer: The recomendation of the reviewer about comparison with results of the literature, is quite valid. However, the literature using the two additives reported in the present study is very scarse or are hard to access, this difficulting further comparison with the present results.

In any case the discussion section was reviewed according to the reviewer recomendation, in order to improve clarity on which was the better treatment.

5. Why is NDE important for the determination of icorr? NDE is the phenomenon of abnormal behavior of Mg and Al, where at high anodic potentials the dissolution of the metal is accompanied by the evolution of hydrogen. Abnormal behavior is also observed in these metals at high cathodic potentials: in addition to the evolution of hydrogen, the metal dissolves in parallel. Therefore, the NDE should have no effect on the quantities determined at the corrosion potential or at the open circuit potential.

Answer: We agree with reviewer when said that “NDE is the phenomenon of abnormal behavior of Mg and Al, where at high anodic potentials the dissolution of the metal is accompanied by the evolution of hydrogen”. However, as was commented in the paper: “The essential feature of the NDE is that the rates of both, the anodic and cathodic reactions, increase with the applied anodic potential, which is unusual (33)”.

The evolution of H2 during anodic polarization take place by the reduction reaction of protons promoted by the Mg-byproducts formed during anodic dissolution, in a not controlled manner. This is the reason why corrosion rate measured by H2 evolution is commonly higher than that measured by polarization technique. Conversely, it is not possible the dissolution of Mg at high cathodic overpotentials, where only the cathodic reaction of H2 evolution is promoted.

  1. As far as I know, a Tafel slope of about 120 mV/dec is characteristic for the hydrogen evolution reaction.

Answer: We agree with the reviewer' comment, considering that at bare Mg a Tafel slope of about 120 mV/dec is commonly found. However, when the substrate is a like-ceramic coating composed by MgO and Mg3(PO4)2 and formed during PEO process the observable Tafel slope is usually larger than 120 mV/dec.

7. Line 282: see references [56]. 

Answer: The reference was added and corrected J. Fischer, M. H. Prosenc, M. Wolff, N. Hort, R. Willumeit, F. Feyerabend, “Interference of magnesium corrosion with tetrazolium-based cytotoxicity assays”, Acta Biomater. 6 (2010), 1813–1823.

  1. Check that the references complies with journal regulations. 

Answer: The references were organized according the journal regulations.

  1. The details of the authors' contributions are missing.

Answer: This was added to the manuscript:

Conceptualization and methodology: M.E.R and  L.F.B; Formal Analysis, M.E.R, L.F.B, S.R, J.A.C, J.G.C, F.E;  writing—original draft preparation and editing, M.E.R, L.F.B, J.A.C, J.G.C, F.E; Funding acquisition, F.E. All authors have read and agreed to the published version of the manuscript.

Reviewer 2 Report

The authors applied plasma electrolytic oxidation to modify the magnesium. The mophology, texture, composition and corrosion resistance were researched in detail. However, some issues should be addressed before publication.

1.       It must have a scaleplate in Figure 5.

2.       How about the accuracy of the corrosion rate? The authors should explain it.

3.       Even though potentiodynamic polarization test is an efficient method to evaluate the corrosion resistance, EIS is more accurate to determine the anti-corrosion performance which is tested under quasi-stable state.

4.       Some important reference should be cited such as: Colloids and Surfaces A: 2023, 661, 130918; Prog. Org. Coat. 2022, 170, 106971.

Author Response

The authors applied plasma electrolytic oxidation to modify the magnesium. The mophology, texture, composition and corrosion resistance were researched in detail. However, some issues should be addressed before publication.

  1. It must have a scaleplate in Figure 5.

Answer: This image was removed due to its low resolution and its little contribution to the discussion of the results.

  1. How about the accuracy of the corrosion rate? The authors should explain it.

Answer: The following explanation of the corrosion rates obtained by hydrogen evolution is added in the text: “Generally, the corrosion rate of Mg alloys obtained from hydrogen evolution is different if compared with other methods frequently used (for example, mass loss). This disagreement was explained by different authors [Kirkland et al, Kappes et al], noting that those calculations did not consider the amount of hydrogen that remains dissolved in the electrolyte. In our experiments, the solution was pre-saturate with H2 to solve the problems related with dissolved H2. Other possible sources of error were negligible or easily avoided, including the variation of gas solubility with temperature and the use of polymeric material permeable to H2 in the measuring setup.”

N.T. Kirkland, N. Birbilis and M.P. Staiger, Assessing the corrosion of biodegradable magnesium implants: A critical review of current methodologies and their limitations, Acta Biomater., 2012, 8, p 925–936.

Kappes, M. Iannuzzi and R. M. Carranza, Hydrogen embrittlement of magnesium and magnesium alloys: A review, J. Electrochem. Soc., 2013, 160, p C168–C178.

Even though potentiodynamic polarization test is an efficient method to evaluate the corrosion resistance, EIS is more accurate to determine the anti-corrosion performance which is tested under quasi-stable state.

The reviewer is right. EIS is more indicated to establish the degree of protection of the anodic layer and therefore the anticorrosive behavior of each sample. Polarization was used with the idea of obtaining the instantaneous corrosion rates and comparing them with the corrosion rates at longer times obtained in the immersion tests. However, discrepancies were found, and this was analyzed in the manuscript.

  1. Some important reference should be cited such as: Colloids and Surfaces A: 2023, 661, 130918; Prog. Org. Coat. 2022, 170, 106971.

Answer: Thank you very much for the suggestion. The second reference was cited in section 3.3.3. However, the first reference was not incorporated because EIS was not included in our manuscript.

Reviewer 3 Report

Dear authors,

I was very interested in your article, but there are a few changes that need to be made:

1. The abstract should be reduced to 200 words. It is also necessary to concretize the results obtained in the abstract. It is also necessary to specify the purpose of this study.

2. In the introduction, to increase readers' interest, it is necessary to write in more detail about the existing types of implants based on titanium alloys and possible surface treatment methods. It is also necessary to compare magnesium alloys and their advantages compared to existing implants. As a recent and interesting publication, I suggest you read the article https://doi.org/10.3390/ma15207374.

3. In paragraph 2.3, it is necessary to add manufacturers of chemical reagents and equipment used.

4. Sec. 2.2 it is necessary to add the range of scanning angles, scanning speed, and type of anode in XRD analysis. Raman analysis needs to add the wavelength and power of the laser used.

5. Section 2.3.2 incorrectly indicates the scanning speed - 1.6 μV/s. It is necessary to specify the type of reference electrode used more precisely. Add a type of counter electrode.

6. P. 3.1 It is incorrect to call the micro-arc oxidation process - anodizing. Please change everything to either micro-arc oxidation or PEO.

7. Line 136 The phase needs to be renamed to the stage. Discussing the coating formation process in more detail and adding literary references is necessary.

8. 5 - Photos of inferior quality. It needs to be corrected.

9. It is necessary to correct wherever it says anodizing on PEO.

10. Reaction 1 is not correct. Correct. Write the complete ionic reaction.

Each statement of the authors must substantiate or add literary references. It is necessary to describe the results of the research in more detail. For example, paragraph 3.4 should be described in more detail and related to the results presented in the article. Need a profound correction of the English language.

Author Response

  1. The abstract should be reduced to 200 words. It is also necessary to concretize the results obtained in the abstract. It is also necessary to specify the purpose of this study.

Answer: The abstract was adapted to the requirements.

  1. In the introduction, to increase readers' interest, it is necessary to write in more detail about the existing types of implants based on titanium alloys and possible surface treatment methods. It is also necessary to compare magnesium alloys and their advantages compared to existing implants. As a recent and interesting publication, I suggest you read the article https://doi.org/10.3390/ma15207374.

Answer: We appreciate the suggestion of the reviewer. The article aimed to use pure Mg and improve its performance to avoid using other elements, such as Al or RE, that may be toxic during degradation. As the objective was also Mg, as permanent implants are not the purpose of this paper we didn´t focus in materials such us Ti.

  1. In paragraph 2.3, it is necessary to add manufacturers of chemical reagents and equipment used.

Answer: The required information was added in section 2.3.

  1. Sec. 2.2 it is necessary to add the range of scanning angles, scanning speed, and type of anode in XRD analysis. Raman analysis needs to add the wavelength and power of the laser used.

Answer: This information was included in section 2.2.

  1. Section 2.3.2 incorrectly indicates the scanning speed - 1.6 μV/s. It is necessary to specify the type of reference electrode used more precisely. Add a type of counter electrode.

Answer: The scanning speed was corrected. The required information about reference and counter electrode was added.

  1. P. 3.1 It is incorrect to call the micro-arc oxidation process - anodizing. Please change everything to either micro-arc oxidation or PEO.

Answer: The text was corrected.

  1. Line 136 The phase needs to be renamed to the stage. Discussing the coating formation process in more detail and adding literary references is necessary.

Answer: The text was corrected, and further description of the process was added.

  1. 5 - Photos of inferior quality. It needs to be corrected.

Answer:This image was removed due to its low resolution and its little contribution to the discussion of the results.

  1. It is necessary to correct wherever it says anodizing on PEO.

Answer: The text was corrected.

  1. Reaction 1 is not correct. Correct. Write the complete ionic reaction.

Answer: This information has been included in the text.

Round 2

Reviewer 1 Report

The revised version of the manuscript has been significantly improved, and the manuscript is now available for publication in Corrosion and Materials Degradation.

Reviewer 2 Report

Accept

Reviewer 3 Report

Good job.